# Antidepressant-like Effects of Renin Inhibitor Aliskiren in an Inflammatory Mouse Model of Depression

**DOI:** 10.3390/brainsci12050655

**Published:** 2022-05-17

**Authors:** Sami I. Alzarea, Hassan H. Alhassan, Abdulaziz I. Alzarea, Ziad H. Al-Oanzi, Muhammad Afzal

**Affiliations:** 1Department of Pharmacology, College of Pharmacy, Jouf University, Sakaka 72341, Aljouf, Saudi Arabia; samisz@ju.edu.sa; 2Department of Clinical Laboratories Sciences, College of Applied Medical Sciences, Jouf University, Sakaka 72341, Aljouf, Saudi Arabia; h.alhasan@ju.edu.sa (H.H.A.); zhaloanzi@ju.edu.sa (Z.H.A.-O.); 3Department of Clinical Pharmacy, College of Pharmacy, Jouf University, Sakaka 72341, Aljouf, Saudi Arabia; aizarea@ju.edu.sa

**Keywords:** lipopolysaccharides, neuroinflammation, depression, Aliskiren, renin inhibitor

## Abstract

Depression is considered a neuropsychic disease that has global prevalence and is associated with disability. The pathophysiology of depression is not well understood; however, emerging evidence has indicated that neuroinflammation could contribute to developing depression symptoms. One of the factors that have a role in the development of neuroinflammation is the renin–angiotensin system. Therefore, the goal of the current study is to determine the antidepressant-like effects of Aliskiren, a renin inhibitor, against lipopolysaccharide (LPS)-induced depressive-like behavior in mice, glial cell activation, and the upregulation of proinflammatory cytokines in the prefrontal cortex. For behavioral studies, the open field test (OFT), tail suspension test (TST), forced swim test (FST), and sucrose preference test (SPT) were used. Inflammatory markers were assessed using real-time polymerase chain reaction (RT-PCR). LPS administration (0.5 mg/kg, intraperitoneal injection (i.p.)) sufficiently reduced the number of crossings in OFT, whereas Aliskiren pretreatment (10 mg/kg, i.p.) attenuated the LPS effect for two hours after LPS injection. The treatments did not show effects on locomotor activity in OFT 24 h after LPS administration. LPS increased the immobility time in TST and FST or reduced sucrose consumption in SPT after 24 h. Aliskiren reversed the effects induced by LPS in TST, FST, and SPT. CD11 b mRNA, a microglial marker, GFAP mRNA, an astroglial marker, and proinflammatory cytokines genes (TNF-α, IL-1β, and IL-6) were upregulated in the prefrontal cortex in LPS exposed animals. However, Aliskiren reduced LPS-induced inflammatory genes in the prefrontal cortex. Hence, the outcomes conclude that Aliskiren prevents depressive illness associated with neuroinflammation in humans.

## 1. Introduction

Depression affects almost 350 million people worldwide. Multiple genetic, as well as environmental factors, are found to cause depression [1]. Depression leads to elevated comorbidities and casualties, inverse health problems, less work productivity, and elevated health care costs [2]. More or less, 0.95 million suicides annually and almost 2950 suicides daily are recorded because of depressive illnesses [3]. The prevalence rates in young adults are very high; 25% are under the age of 19 years only. Reports are alarming since about 40% of the patients show therapeutic failure to existing pharmacological treatments [4]. Researchers still cannot find definitive pathophysiological mechanisms. Modern investigations are focused on enlightening disease pathogenesis and advancing therapeutic inefficiency. Recently, inflammation was suggested as one of the important factors in depression. Elevated peripheral proinflammatory intermediaries are consistently reported in depression and in many other mood disorders as well [5]. Altered neuroinflammation and hyperactivated microglial cells are found to play a critical role in the pathology of depression [6,7,8,9]. The severity of illness is graded by measuring altered levels of proinflammatory intermediaries, especially interleukins (IL-1β and IL-6), tumor necrosis factor-alpha (TNF-α), and interferon-gamma (IFN-γ) [10]. Once the protective barrier to the brain (the blood–brain barrier) and choroid plexuses are compromised, the microglial cells become activated and, worse, inflammation occurs via the release of proinflammatory mediators [11].

Lipopolysaccharides (LPS) (bacterial endotoxins) are experimentally proved to evoke microglial cells and acute phases of inflammation, which cause depression in humans and depressive behavior in rodents [12]. LPS interacts with TLR4 receptors on microglial cells and evokes an immunological cascade of reactions. LPS–TLR4 interaction stimulates nuclear factor kappa –B (NFk-B) and mitogen-activated protein kinase (MAPK) signaling pathways [13] and triggers inflammatory mediator release, mainly tumor necrosis factor-α and Interleukins (IL-1β, IL-6) [14]. This also upregulates the expressions of Inducible nitric oxide synthase (iNOS) and Cyclooxygenase-2 (COX2), which further aggravate neuroinflammation [15]. Minocycline is a well-known antibiotic with anti-inflammatory, immunomodulatory, and neuroprotective effects. It was proven scientifically that inhibitory effects on the activities of key enzymes, such as the iNOS [16] inhibition of caspase-1 and caspase-3 activation [17] and anti-apoptotic effects [18,19,20], cause the reduction of p38 MAPK phosphorylation [21]. Thus, it was shown that it also inhibits microglial polarization and these effects could also be ascribed to its activity on neurons. In addition, microglia by a selective inhibitor, minocycline, inhibits neuroinflammation and controls microglial activities, making it a good target for the treatment of depression [22].

The general or classical renin–angiotensin system (RAS) is the endocrine league accountable for controlling various bodily and pathologic developments, such as hypertension, body water and electrolytic balance, and inflammation [23]. However, secondary RAS (local RAS) has been documented in numerous tissues, together with the brain. Local angiotensin plays a pivotal role in pathological modifications paracrinally and modifies immune cells. The paracrine RAS has been documented to regulate microglial polarization. The brain RAS has been well documented to play an important role in neuroinflammation via oxidative free radical generation and the release of various potentially aggressive inflammatory cytokines [24]. By considering RAS as a significant target, researchers explored its role in depression and claimed it could be an antidepressant via its anti-inflammatory mechanisms [25,26]. NADPH-oxidase (Nox) activation in microglial cells controls the shift between proinflammatory and immunological morphologic types. Nox stimulation endorses proinflammatory reactions and obstructs the immunoregulatory phenotype of microglial cells [27].

Inhibition of the enzymatic activity of renin activity, the rate-limiting phase of RAS, is well known to encourage inflammation, vasoconstriction, reactive free oxygen radicals generation, trigger fibrosis via various chemokines, and cytokine-mediated inflammatory courses in the tissue [28]. Aliskiren (renin inhibitor) was the first to touch the market. Hence, via an intense literature survey related to Aliskiren, LPS and its various manifestations, with inflammation and depression as its long-lasting effect, we planned to study the effect of RAS and its influence on an inflammatory mouse model of depression induced by LPS administration.

## 2. Material and Methods

### 2.1. Animals

Adult Wistar albino mice (20–25 g) were selected randomly and accommodated for one week in ambient laboratory surroundings, which comprised a module of 12 h (light: dark) cycle, 22 ± 2 °C temperature, and an average 55% humidity with a deliberate approach to normal rodent chow and water. The Experimental protocol was approved by the Local Committee of Bioethics at Jouf University, Sakaka, Aljouf, Saudi Arabia (6-02-43), and the experiments were carried out at the College of Pharmacy, Jouf University, Sakaka, Aljouf, Saudi Arabia.

### 2.2. Chemicals

Aliskiren and LPS (*Escherichia coli*, serotype 0127:B8) were purchased from Toronto Research Chemicals Inc. (North York, ON, Canada).

### 2.3. Study Design

The animals were selected and divided into eight groups (*n* = 8). All experimental animals were administered with Aliskiren (1, 3, 10 mg/kg; intraperitoneal injection (i.p.)) on the first day at ‘0’ h. Twenty-four (24) hours after the first dose, the second dose of Aliskiren was administered. On the third day, 24 h after the second dose, animals were administered with the third dose of Aliskiren and simultaneously administered with LPS (0.5 mg/kg; i.p.) [29]. Then, after two hours of LPS administration, locomotor activity was performed. Then, from ‘0’ hours to 24 h, the sucrose preference test (SPT) was performed (for 24 h). Twenty-four (24) hours after LPS administration, OFT was performed again to observe the sickness behavior. A tail suspension test (TST) was performed 25 h after LPS administration, and a forced swim test (FST) was conducted 26 h after LPS administration. Twenty-seven (27) hours after LPS, animals were sacrificed by the decapitation method, and tissues were isolated for further intracellular estimations of various genes. (Figure 1).

### 2.4. Open-Field Test (OFT)

For the assessment of locomotor activity as an indicator of LPS-induced sickness behavior, we performed OFT [30]. Before the conduction of experiments, animals were acclimatized to the test room for at least an hour. The apparatus used was comprised of a plexiglass square box (50 cm × 50 cm × 40 cm). The floor was divided into 25 equal parts (10 cm × 10 cm) marked with lines. Each mouse was placed in a small square (10 cm × 10 cm) and permitted to explore independently for six minutes. The total number of squares crossed with its four paws was completely deliberated as locomotion. The equipment was cleaned after the test in order to avoid any influence by fall orders of urine and feces of the animals used in earlier experiments.

### 2.5. Tail Suspension Test (TST)

TST was conducted as per the earlier established method [30]. Briefly, mice were isolated in an acoustic and visually isolated environment. They were held 50 cm above the floor by sticking about 1 cm of adhesive tape at the tip of the tail individually. After 120–180 s of characteristic vigorous motions (struggling activities, consistent efforts to catch the adhesive tape, body rotations, spinning, or jerking), the mice hung and became stationary. Immobility is defined as the absenteeism of any movement associated with the limbs or body except those of respiration. Reduction in immobility indicates antidepressant-like influences in the total immobility duration in seconds, which was documented manually during the six minute session using a stopwatch.

### 2.6. Forced Swim Test (FST)

FST was used to study antidepressant-like activity in rodents. It was carried out as described [30]. Concisely, in a tube-shaped plexiglass chamber (45 cm high × 20 cm diameter) filled with 25 cm of water (25 ± 1 °C), mice were permitted to swim for six minutes. In the course, a video was recorded and scrutinized to document immobilization time. Immobility was projected when no surplus activities were observed, except those that required keeping the head above the level of the water. Mice were removed from the cylinder instantly after the test, dried using paper towels, and brought back to their cages.

### 2.7. Sucrose Preference Test (SPT)

SPT was established to assess anhedonic behavior in rodents [22]. Mice were housed separately and independently. They were provided with free access to two bottles containing 1% sucrose solution and normal water after LPS injection for 24 h. The weight of the mice was recorded before beginning the experiment. Twenty-four (24) hours after LPS administration, the bottles were removed and weighed to determine sucrose consumption. The consumption was calculated based on the following equation: sucrose preference % = (sucrose bottle weight (g)/sucrose bottle weight (g) + water bottle weight (g)) × 100.

### 2.8. Real-Time PCR Analysis

After animal scarification, brains were collected, and the prefrontal cortices were immediately isolated. The isolated tissues were stored at −80 °C until RNA extraction. Total RNA was extracted from the tissues using TRIzol^®^ (Invitrogen, Waltham, MA, USA) based on the manufacturer’s instructions. A high-Capacity cDNA Reverse Transcription Kit (Applied Biosystems, Foster City, CA, USA) and Master Cycler Personal (Eppendorf, Enfield, CT, USA) were used to generate cDNA from the extracted RNA samples. For the relative quantification of mRNA levels, a quantitative polymerase chain reaction (qPCR) was performed on the StepOnePlus quantitative real-time PCR system (Applied Biosystems). An SYBR green PCR kit (Applied Biosystems) was used in the assay along with specific primers (Integrated DNA Technologies, Coralville, IA, USA). For the quantification of target gene expression, the ΔΔ Cq method was used relative to the housekeeping gene, GAPDH (Table 1).

### 2.9. Statistical Analysis

The results were expressed as means ± SD and analyzed using GraphPad Prism software. The differences between control groups and LPS groups were determined by two-way ANOVA followed by Tukey’s test for post hoc comparisons. The significant difference among the groups was considered when the *p*-value was less than 0.05. 

## 3. Results

### 3.1. Effect of Aliskiren on LPS-Induced Sickness Behavior in Mice

#### Open Field Test

The assessment of locomotor activity of Aliskiren in the LPS-induced model was conducted by performing four different behavioral tests. Results of the OFT conclude that two hours after LPS injection, experimental animals (which were treated with Aliskiren prophylactically at 1 mg/kg, 3 mg/kg, and 10 mg/kg doses) showed sickness behavior. The experimental animals had significantly decreased (* *p* < 0.05) open-field exploration after LPS injection, while the Aliskiren-treated animals showed decreased sickness behavior (* *p* < 0.05) (Figure 2A).

Similarly, in the test performed after 24 h of LPS administration, the test results depict that the sickness behavior was resolved after 24 h (Figure 2B). This test was performed to study the sickness behavior in animals after LPS administration. The result of the experiment suggests the development of sickness behavior in animals after two hours of LPS administration.

### 3.2. Effect of Aliskiren on LPS-Induced Depressive-like Behavior in Mice in TST, FST and SPT

#### 3.2.1. Tail Suspension Test

After 25 h of LPS administration in mice, which were prophylactically treated with Aliskiren, three doses of Aliskiren (each after every 24 h) at three dose levels (1 mg/kg, 3 mg/kg, and 10 mg/kg, respectively), the TST was performed. The results of the TST suggest that animals treated with Aliskiren (1 mg/kg, 3 mg/kg, and 10 mg/kg) showed differences in their immobility times after LPS administrations. The immobility times were found to decrease in the mice pretreated with Aliskiren. Experimental mice treated with 1 mg/kg, 3 mg/kg, and 10 mg/kg of Aliskiren showed highly significant (** *p* < 0.01) decrements in immobility times (Figure 3A) as compared to the controls. The findings also suggest that the immobility times were decreased in animals significantly (* *p* < 0.05) after the treatments. When the results were compared with saline only animal groups, highly significant results were achieved at 10 mg/kg of Aliskiren (Figure 3A). This test was performed to study the depressive-like behavior in animals after 25 h of LPS administration. The result of the experiment suggests the development of depressive-like behavior after sickness behavior in animals (Figure 3A).

#### 3.2.2. Forced Swim Test

Similarly, the findings of the FST performed after 26 h of LPS exposure suggest that the animals had depressive-like behavior because they had significantly longer immobility periods (* *p* < 0.05) as compared to the controls, while the treated animals had significantly low immobility periods as compared to LPS (* *p* < 0.05). The animals treated with a high dose of Aliskiren (10 mg/kg) showed more significant results (Figure 3B).

#### 3.2.3. Sucrose Preference Test

This test was performed to study anhedonia-like behavior in experimental mice. The results of the test performed indicate that control animals provided access to the sucrose solution, and the sucrose preference was noted to be significantly higher (** *p* < 0.01) as compared to the animals which were exposed to LPS. The animals administered with LPS showed a significantly low (** *p* < 0.01) preference for sucrose solution. Observations also suggested that Aliskiren treatment to the animals increased the pleasurable effects and declined the depressive-like behavior in the rodents dose-dependently. Higher significant results were observed in the experimental group which was treated with 10 mg/kg of Aliskiren (Figure 3C).

### 3.3. Assessment of Aliskiren Effect on Glial Cell Marker in Prefrontal Cortex

The RT-PCR assay was conducted to estimate the effect of LPS on astrocytes and microglial cells. The specific marker for microglial cells is CD11b, while the one for astrocytes is glial cell acidic protein (GFAP). Findings of the research protocol suggest that after 24 h of LPS administration, the level of mRNA expression of CD11b was significantly high (* *p* < 0.05) in the animals as compared to controls, while the treatment with Aliskiren decreased these elevated levels significantly (* *p* < 0.05) in the groups which were treated prophylactically with Aliskiren as compared to animals treated with saline. The levels significantly decreased in animals which were treated with 10 mg/kg of Aliskiren (* *p* < 0.05) (Figure 4A).

A further mRNA expression assay to estimate the GFAP levels from astrocytes depicted the higher values (** *p* < 0.01) in the animals exposed to LPS compared to normal controls. Prophylactic treatment with Aliskiren decreased these elevations dose-dependently, and higher decrements (* *p* < 0.05) were recorded in animals with 10 mg/kg doses (Figure 4B).

### 3.4. Assessment of LPS and Aliskiren Effect on Proinflammatory Cytokines in Prefrontal Cortex

#### 3.4.1. Effect of LPS and Aliskiren on TNF-α

The RT-PCR of tissue homogenates of the prefrontal cortex suggests that after the administration of LPS, the levels of TNF-α mRNA increased significantly (** *p* < 0.01) as compared to normal control animals. Prophylactic treatment of Aliskiren significantly inhibited the generation of TNF-α mRNA (* *p* < 0.05) after LPS exposure in a dose-dependent manner. Highly significant inhibition (* *p* < 0.05) in the levels of TNF-α mRNA was noted in the animals which were treated with 10 mg/kg of Aliskiren (Figure 5A).

#### 3.4.2. Effect of LPS and Aliskiren on Brain IL-1β Levels

Tissue homogenates of the prefrontal cortex were analyzed quantitatively for the estimation of IL-1β gene expressions. The results suggest that the level of IL-1β mRNA had a significantly high increase (*** *p* < 0.001) in animals which were exposed to LPS and prophylactically treated only with the vehicle as compared to normal controls. The treatment of animals with Aliskiren inhibited the generation of IL-1β mRNA in the prefrontal cortex in a dose-dependent manner, and higher significant inhibition (*** *p* < 0.001) was recorded with the 10 mg/kg of Aliskiren dose as compared to saline only treatment (Figure 5B).

#### 3.4.3. Effect of LPS and Aliskiren on Brain IL-6 Levels

Analytical assays for the assessment of IL-6 mRNA levels in tissue homogenates exhibited significant upregulation (* *p* < 0.05) in experimental groups which were exposed to LPS as compared to normal controls. Treatment with an Aliskiren dose prevented the generation of proinflammatory cytokine IL-6 in a dose-dependent manner. The most significant inhibition (* *p* < 0.05) was noted in experimental animals, which were treated with a 10 mg/kg dose (Figure 5C).

## 4. Discussion

Lipopolysaccharides (an immunologic, cell wall component of Gram-negative bacteria) stimulate Toll-like receptors (TLR-4) present on the microglial cell surface [31,32]. The stimulation of TLR-4 receptors causes the release of inflammatory mediators, which contribute to inflammation and neuronal injuries chronically [33,34]. The available scientific data exhibit the role of neuroinflammation in neurological diseases [35]. Neuroinflammation can then be preceded by Alzheimer’s disease [36]. Scientific reports also documented the role of neuroinflammation in Parkinson’s disease, Huntington’s disease, and amyotrophic lateral sclerosis. Various preclinical and experimental studies claimed the role of LPS in neuroinflammation [37]. Inflammatory mediators such as TNF-α, IFN-γ, and interleukins (IL1-β, IL-6, IL-12) not only encourage neuroinflammation but also openly distress nerve transmission, causing behavioral mood changes [38]. Experimental procedures conducted to study the influence of LPS administration in rodents confirmed the development of depression, anhedonia, low interest in social activities, fatigue, psychomotor slowing, and cognitive alterations [39]. The RAS is a vascular enzymatic cascade of reactions known to regulate blood volume and pressure [40]. RAS inhibitors such as renin inhibitors, angiotensin-converting enzyme inhibitors (ACEIs), and Angiotensin receptor blockers (ARBs) are therapeutically indicated to treat hypertension. Recent research and development work also suggest their potential roles in neuroprotection [41]. ARBs were neuroprotective in cerebral ischemic and hemorrhagic conditions via the modulation of BDNF/TrkB signaling cascades and suppressing oxidative stress and proinflammatory mediators [42]. Furthermore, some other research demonstrates the strong involvement of RAS insult in dementia, Alzheimer’s disease, and neurodegeneration [43].

LPS administration to animals was documented to cause sickness behavior in experimental animals, which confirms the persistence of sickness behavior for six hours that finally resolves after 24 h [30]. Similarly, the findings of the current research protocol demonstrated sickness behavior after two hours of post-exposure to LPS in experimental animals. The results of the OFT performed after 24 h of LPS administration demonstrated the disappearance of sickness behavior. Pretreatment with Aliskiren significantly reduced the sickness behavior in the experimental animals after two hours of LPS administration. Earlier research reports depicted that LPS administration to experimental mice induces depressive behavior, which is evident by increased immobility times in TST and FST, and a decreased percentage in sucrose preference in SPT [22,30]. The outcome of the current research protocol attested to these findings and demonstrated a significant increase in immobility times in TST and FST, which were performed after 25 h and 26 h of LPS administration. The findings are also in agreement with the results of SPT, and the sucrose preference percentage was significantly reduced. Prophylactic administration of Aliskiren exhibited a significant reduction in immobility times in the TST and FST and increased the percentage of sucrose preference in SPT. More significant results were noted at 10 mg/kg of Aliskiren. LPS contributes to neuroinflammation, which is evidenced by the earlier findings that LPS administration results in microglial cell activation in the brain [44]. Microglial cell activation was linked to the generation of proinflammatory cytokines and was found to play an important role in the pathophysiology of depression [8]. The microglial cell activation marker is CD11b [45], and GFAP is the biomarker for astrocyte stimulation which is induced due to brain injuries or stress to the central nervous system [46]. The present research protocol outcomes exhibited the same. LPS administration caused a significant increase in CD11b levels in prefrontal tissues homogenates, and the same was found with GFAP. These results attest to the earlier findings and the role of LPS in brain injuries and microglial cell activation. Pretreatment with Aliskiren restored these altered levels of CD11b and GFAP in treated animals. Activated microglial cells were very well studied and postulated to increase the levels of proinflammatory cytokines [47], TNF-α, IL-β, IL-6, prostaglandins, nitric oxide, and reactive oxygen species [48,49]. Several other research reports also claimed the influence of LPS on microglia activation, which consequently increases the expressions of proinflammatory cytokines [50]. The current research outcomes were consistent with the earlier reports; LPS administration to experimental animals significantly enhanced the expressions of TNF-α, IL-β, and IL-6, while pretreatment with Aliskiren restored these proinflammatory cytokine levels toward normal values. The effect of the Aliskiren was more significant at a 10 mg/kg dose.

## 5. Conclusions

The results of the research suggest that LPS exposure to experimental animals is an established toxin that causes neuronal inflammation and changes in behavior. LPS-led inflammatory reactions contribute to the development of depression. The findings of the experiment conclude and attest to the earlier reports regarding the involvement of the renin–angiotensin system in neuroinflammatory mechanisms and progression to affective disorders. Aliskiren, a renin inhibitor, is proven to be an alternative for the treatment of neuroinflammation by suppressing microglial cell activation and proinflammatory cytokine generation, which prevent the development of depression. 

## Figures and Tables

**Figure 1 brainsci-12-00655-f001:**
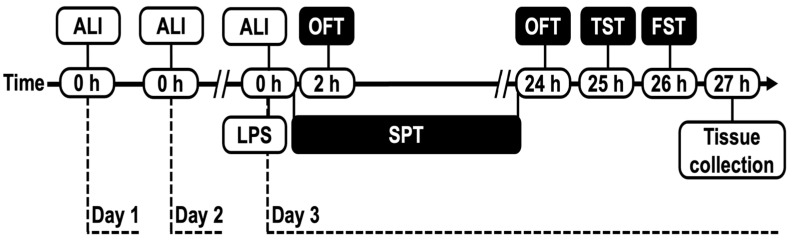
Experimental design.

**Figure 2 brainsci-12-00655-f002:**
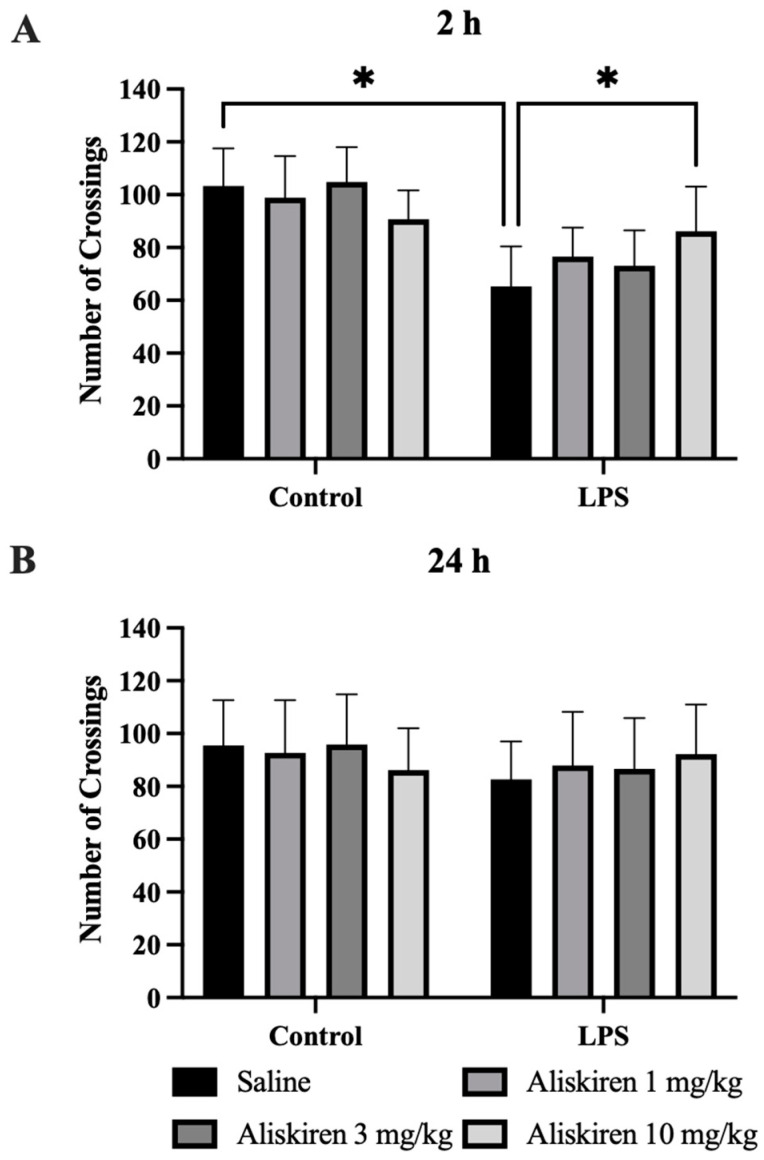
Assessment of Aliskiren Effects against LPS-Induced Sickness Behavior in Mice. (**A**) The effects of Aliskiren on the number of crossings in OFT two hours after LPS administration. (**B**) The effects of Aliskiren on the number of crossings in OFT 24 h after LPS administration. *n* ≥ 6 mice/group. * *p* < 0.05.

**Figure 3 brainsci-12-00655-f003:**
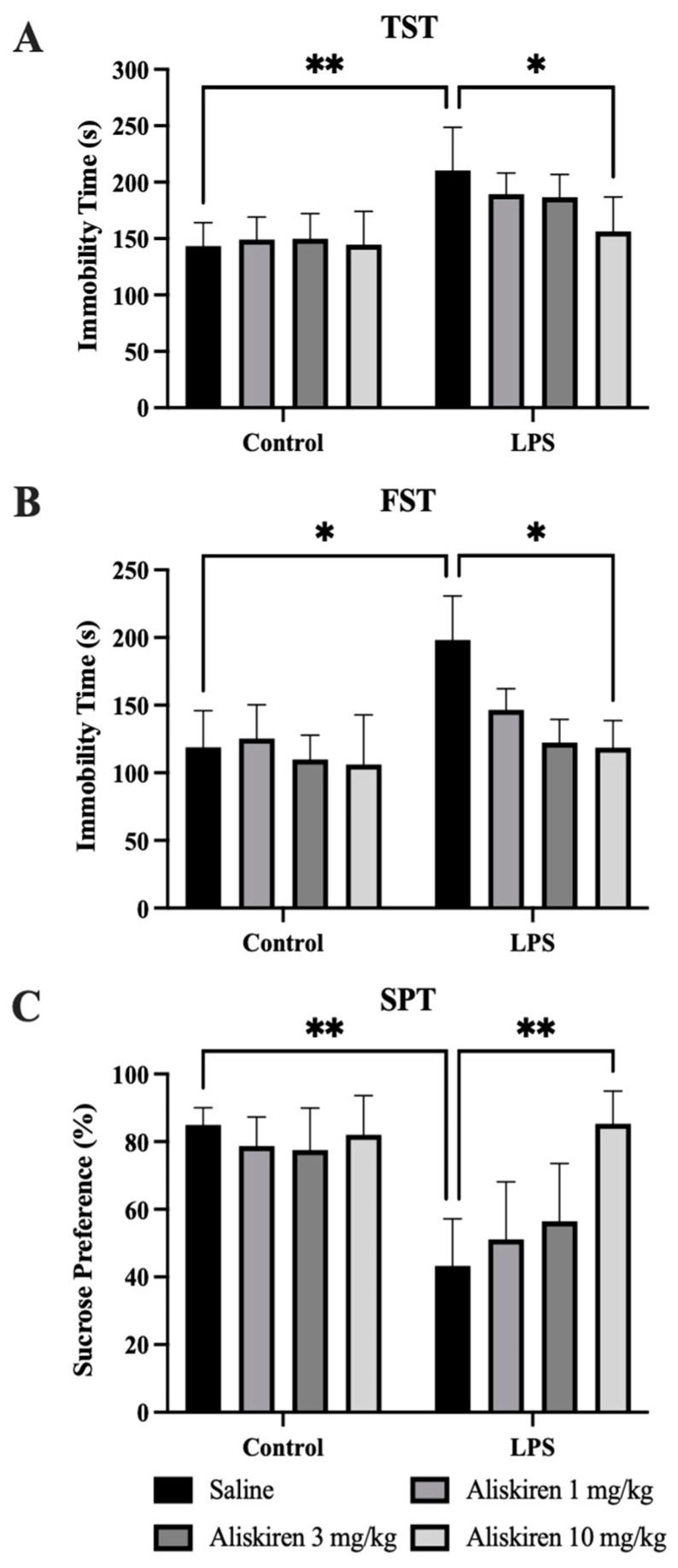
Assessment of Antidepressant-like Effect of Aliskiren in Mice. (**A**) The effects of Aliskiren on immobility time in TST. (**B**) The effects of Aliskiren on immobility time in FST. (**C**) The effects of Aliskiren on sucrose preference % in SPT. *n* ≥ 6 mice/group. * *p* < 0.05; ** *p* < 0.01.

**Figure 4 brainsci-12-00655-f004:**
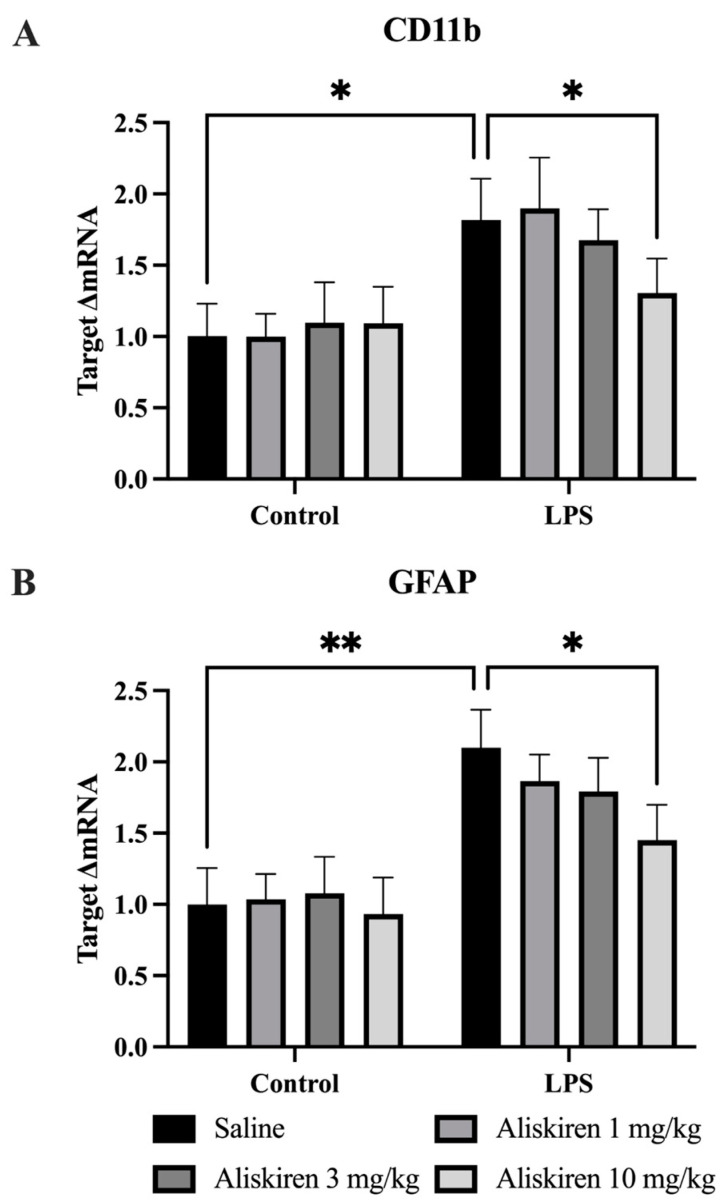
Assessment of Aliskiren Effects on Glial Markers Activation in Prefrontal Cortex. (**A**) The effects of Aliskiren on LPS-induced upregulation of CD11b. (**B**) The effects of Aliskiren on LPS-induced upregulation of GFAP. *n* ≥ 6 mice/group. * *p* < 0.05; ** *p* < 0.01.

**Figure 5 brainsci-12-00655-f005:**
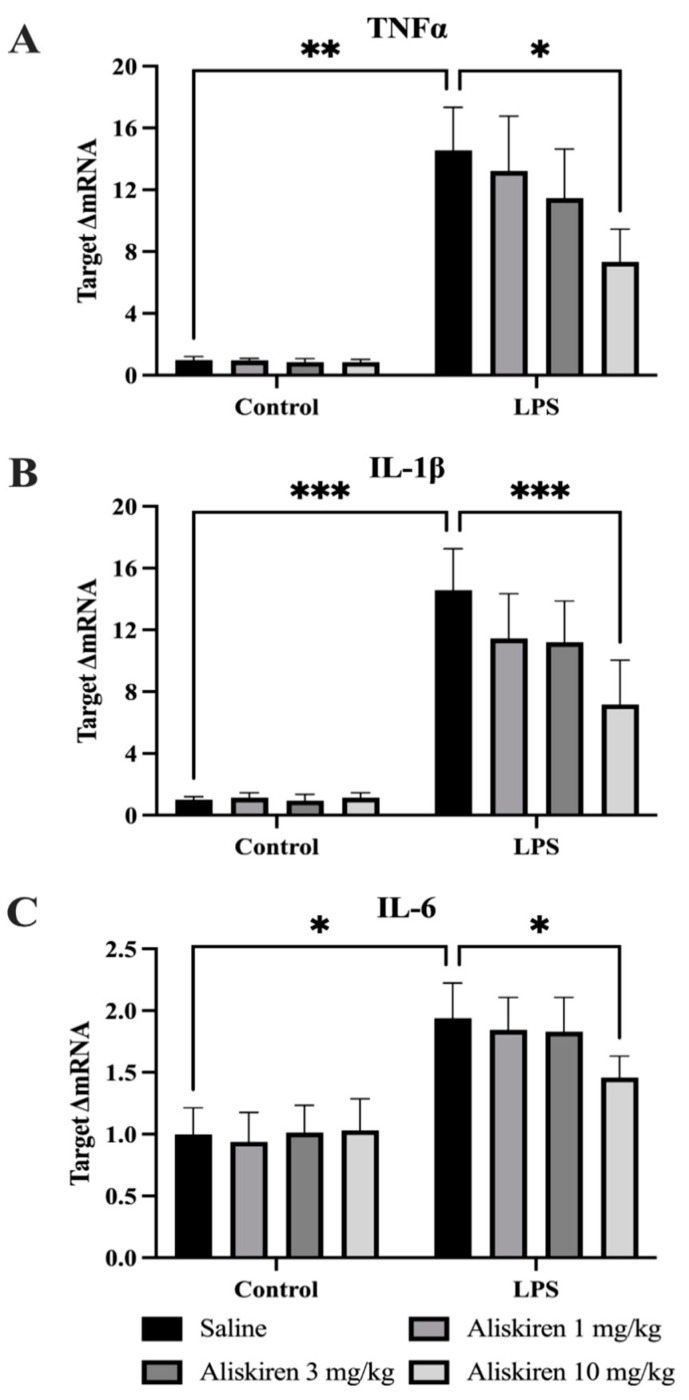
Assessment of Aliskiren Effects on Proinflammatory Cytokines in Prefrontal Cortex. (**A**) The effects of Aliskiren on LPS-induced upregulation of TNFα. (**B**) The effects of Aliskiren on LPS-induced upregulation of IL-1β. (**C**) The effects of Aliskiren on LPS-induced upregulation of IL-6. *n* ≥ 6 mice/group. * *p* < 0.05; ** *p* < 0.01; *** *p* < 0.001.

**Table 1 brainsci-12-00655-t001:** Genes and primers for qRT-PCR (5′–3′).

Gene	Forward	Reverse
GAPDH	GTGGAGTCATACTGGAACATGTA	AATGGTGAAGGTCGGTGT
CD11b	TGTCCAGATTGAAGCCATG	CCACAGTTCACACTTCTTTCA
GFAP	GCATCTCCACAGTCTTTACC	AACCGCATCACCATTCCT
TNF-α	TCTTTGAGATCCATGCCGTT	AGACCCTCACACTCAGATC
IL-1β	CTCTTGTTGATGTGCTGCT	GACCTGTTTGAAGTTGAC
IL-6	GAGGATACCACTCCCAACAGACC	AAGTGCATCATCGTTGTTCATACA

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
