# Peer review of "Antidepressant-like Effects of Renin Inhibitor Aliskiren in an Inflammatory Mouse Model of Depression"

_brainsci, 2022, doi:10.3390/brainsci12050655_

Round 1

Reviewer 1 Report

The manuscript describes an experiment aimed at evaluating the antidepressant efficacy of a renin inhibitor. The authors demonstrate that pre-treatment with aliskiren attenuates LPS induced sickness behavior (e.g., line crossing at 2 hours), depressive-like behavior at just over 24 h post LPS, and brain qPCR markers of inflammation (e.g., 27 h post?).  While inflammation-induced depressive-like behavior is an important topic of investigation, I have a number of concerns about the manuscript in its current form.

  1. Extensive grammatical editing is required.
  2. There are a number of other studies that consider the role of the renin-angiotensin system in inflammation and/or depression that should be incorporated into the introduction and/or discussion.
  3. As a more minor notes, please use standard accepted guidelines for formatting gene names and also consider shifting error bars from SD to SEM. 
  4. A timeline clarifying the experiment would be helpful. 
  5. Additionally, I wonder if the authors can comment on the full reversal of behavior with a much more marginal reduction in frontal cortex Tnf and Il1b?
  6. Finally, as this is a prevention model rather than a treatment model and given that this drug is associated with blunting inflammation itself, the observed effects could be directly through preventing LPS-induced inflammation. As such, I find claims related to treatment potential of the agent to be premature.  If the authors could comment more about where the envision the actions of aliskiren to be in the cascade of LPS-induced depression this would be helpful.  

Reviewer 2 Report

In the present article, the authors explore the antidepressant potential of aliskiren, a renin inhibitor, in neuroinflammatory model of depression. The idea is novel and interesting. However, the article needs to be improved. Here are the comments.

General comments:

  1. The article needs considerable improvement in the English language. Numerous sentences are not grammatically correct and some sentences are hard to understand.
  2. It would be good if the results for the CD11b and GFAP would be confirmed by immunohistochemistry and/or Western blot.

  1. Abbreviations should be introduced when words are mentioned for the first time. After that, the abbreviations should be used and not the full words (e.g. LPS line 50)
  2. The name of the drag aliskiren is written in small letters.

Abstract

  1. Please correct the English in the first and the last sentences of the abstract.

Introduction:

  1. Please, find more recent and more adequate references about epidemiological data regarding depression.
  2. Sentences in lines 36-38 and 38-40 should be divided in two, for more clarity.
  3. Please, provide more background and references about LPS-induced depression (e.g. O'Connor, 2009…)
  4. Please, give some more background about involvement of RAS in depression and that its blockade could have beneficial effects.
  5. Please rewrite the sentence in lines 65-67, it is not clear and correct - the activation of RAS, not its inhibition, is involved in inflammation.
  6. The last paragraph should end with the aim of the research and how the authors intended to investigate that.

Methods:

  1. The section Study design should be written more clearly and grammatically correct. The authors should provide the exact number of animals per group. Additionally, it would be nice if the authors provide the scheme of their experiment in order to better understand what were the groups, treatments and the timeline of experiments.
  2. In the section Open field test, please state that this test is used for assessment of locomotor activity as an indicator of LPS-induced sickness behavior.
  3. In the section Tail suspension test, lines 110-111 use the word “immobility” not “immobilization”. The sentence in lines 112-114 should be divided into two.
  4. It is not clear when the SPT test started and when it ended.
  5. In the section Real time PCR, please give primer sequences that were used. Also, give the full words for abbreviation “qPCR” (line 137).
  6. In the section Statistical Analysis, please state that the two-way ANOVA was used to assess the effects of LPS and aliskiren, while the differences between the groups were analysed by post-hoc test.

Results:

  1. F and p values from ANOVA should be stated for each result, i.e. main effects of factors LPS and aliskiren, as well as their interaction, if it is significant. Separately, p values from post-hoc test should be stated.
  2. Generally, all sections in the Results should be rewritten to be clearer and grammatically correct. Also, always use past test when reporting results.
  3. In the figure legends, it should be stated from which statistical test p values were represented.
  4. OFT is not a test for depressive-like behaviour but for locomotor activity, which can be a confounding factor in assessing depressive-like behaviour in TST and FST. Please, correct that.
  5. In the line 155, you state “treated animals showed decreased sickness behaviour”, but you should state more clearly that it was aliskiren treatment.
  6. In the line 195, words “disinterest in social act” do not belong to the sentence.
  7. In the Results, it is sufficient to briefly state that GFAP is assessed as a marker of astrocytes, while CD11b+ a marker of microglia and infiltrating leukocytes. Further elaboration about the usage of these markers should be left for discussion.
  8. In the Figure 3., there are no asterisks in the pictures.
  9. In the line 229, please write “dose-dependent” instead of “dose dependent”.
  10. In the line 245, please write “Effect of LPS and aliskiren on brain IL-6 levels” instead of “Effect of aliskiren and LPS on brain IL-6 levels” in order to standardise all titles.

Discussion:

  1. The discussion should be rewritten, extended and more focused on neuroinflammatory model of depression and not neurological diseases. It should start with a brief summary of the main findings of the article and then be interpreted in the light of available literature. The sentences should be clearer. The results regarding the effects of LPS and aliskiren on behaviour and molecular analyses should be divided into separate paragraphs. Also, you could discuss what would be the therapeutical potential of prophylactic treatment by aliskiren.
  2. Several sentences are not clear and grammatically correct (e.g. in the lines 273-275).
  3. Please, use the phrase “sickness behaviour”, instead of “sickness-like behaviour”.

Conclusion:

  1. The conclusion is not adequate since the model that was studied was LPS-induced depression, not brain injury. Please, rewrite the conclusion accordingly.

Reviewer 3 Report

In this manuscript, Alzarea and colleagues explore the antidepressant effects of the renin inhibitor Aliskiren in a LPS-induced model of depression. Whereas the topic might be of interest, an extensive revision needs to be performed for 1) English language and 2) statistics employed. See my specific comments below:

Introduction:

  1. Page 2, line 50-51: is LPS currently being given to patients?
  2. Line 53: minocycline is not a selective inhibitor of microglia.
  3. The introduction does not provide the necessary information to understand the rational of this story. First, whereas it is true that elevated peripheral cytokines have been found in the blood of patients with depression, what is the link to neuroinflammation (in other words, the authors should briefly introduce how peripheral inflammation propagates to the bran, important for their peripheral LPS challenge). Are there evidence of central inflammation in depression (for instance TSPO imaging)? If so, this should be mentioned in the introduction.
  4. The introductory part concerning the RAS system is barely understandable (this manuscript need to be revised by a native speaker). The link between RAS and Nox is not mentioned.

Material and methods:

  1. Sex of the mice should be clearly mentioned.
  2. More information about the type LPS used (Bacteria, strain, what type of purification) should given as this can have a huge impact on the inflammatory reaction. Also, why did they chose a dose of 0.5 mg/kg?
  3. The exact number of mice should be given for each experiment.
  4. The experimental approach (which, for clarification, could also be schematized in a figure) is very difficult to read and understand. Is the control group injected with saline? Is the sucrose test performed between “0” hours and 24 hours, thus 2 days before LPS injection (as mentioned in lines 90-91)? Unlike stated at pag.2, lines 85-86, the authors do not inject all the mice with Aliskiren, but also a control group with saline.
  5. Why was the prefrontal cortex chosen?
  6. The authors state that they used a 2-way ANOVA. Is this appropriate? Did they check the normal distribution of their data? Also, they have more than 2 variables: LPS, treatment and dose of treatment. They should use appropriate statistical analysis, otherwise this paper cannot be considered for publication as I cannot evaluate whether the effects they reposted are significant or not. A solution could also be to only focus on 1 treatment dose (10 mg/kg), which is the only one that seems to have an effect from their data.

Results:

  1. The statistical power is not clear. As the authors used a 2-way ANOVA, the main effects (LPS and treatment effect, plus eventual interaction), should be clearly reported. It is unclear whether the significant differences are the results of the post-hoc comparison.
  2. Additional experiments required: Survival curve, peripheral levels of inflammatory cytokines, temperature and body weight (sickness behavior). Moreover, the authors should also check or at least consider the stress induced to the mice by performing the forced swim and the tail suspension test 1 hour apart. CORT levels should be checked.
  3. Figure 3 and figure 4: the stars indicating significant differences are missing.

Reviewer 4 Report

The manuscript entitled “Antidepressant-like effects of renin inhibitor aliskiren in inflammatory mouse model of depression” by Alzarea and coworkers is focusing a very important and actual topic. The basic idea is original and directed toward the effects of aliskiren administration in an inflammatory mouse model of depression. The manuscript is well organized, concise, and covered by appropriate references but still needs to be improved by employing some more specific literature data.

Still, there are some concerns:

The number of animals per group must be indicated.

What software is used for behavioral analyses?

If you take into consideration that:

- an early response (in 2h) can rather be attributed to an acute (anxiogenic) reaction to traumatic manipulation than to behavioral alteration (sickness behavior?);

- the OF test should not be repeated so frequently (two times in 24 hours?);

- OF test is not specific for depression evaluation, and,

- there is quite enough testing suitable for the estimation of depression levels;

therefore, I strongly suggest the authors exclude the results of this test from the manuscript.

The authors should include in the Discussion section the results obtained on the same species, following the same induction of behavioral alterations that described the precise mechanism of LPS-induced behavioral changes due to the fact that it offers numerous relevant elements including cytokines profile, BDNF levels, GABAergic receptors alterations (in the hippocampus), and the impact of TLR-4 receptors (10.1016/j.bbi.2019.01.019).

Round 2

Reviewer 3 Report

Query 2. Line 53: minocycline is not a selective inhibitor of microglia.

Reply: Authors appreciate critical evaluation of manuscript and recommendation to improve the manuscript unto its excellent form. On the basis of previous research reports (K. Kobayashi and colleagues, 2013, cell death and disease, 4, e525) it has been conferred undoubtedly that minocyclin is selective inhibitor of microglia and prevent proinflammatory cascade of reaction.

Re-reply: For clarification, minocycline is a well-known antibiotic with anti-inflammatory, immunomodulatory and neuroprotective effects. The latters include inhibitory effects on the activities of key enzymes, like iNOS (Amin et al., 1997 ), inhibition of caspase-1 and caspase-3 activation (Chen et al., 2000), anti-apoptotic effects (Wang et al., 2003; Domercq and Matute, 2004; Jordan et al., 2007), reduction of p38 MAPK phosphorylation (Corbacella et al., 2004), etc. Thus, although it has been shown that it also inhibits microglial polarization (a concept not accepted anymore in the field as microglia can exist in a variety of cell states), this effects could also be ascribed to its activity on neurons (which obviously might feed-back to microglia). 

Query 7: More information about the type LPS used (Bacteria, strain, what type of purification) should given as this can have a huge impact on the inflammatory reaction. Also, why did they chose a dose of 0.5 mg/kg?

Reply: Information about the LPS strain used has been provided in the manuscript and authors selected 0.5mg/kg dose of LPS on the basis of previous reports

Re-reply: The previous reports used to choose the LPS dose should be clearly stated in the manuscript.

Query 11: The authors state that they used a 2-way ANOVA. Is this appropriate? Did they check the normal distribution of their data? Also, they have more than 2 variables: LPS, treatment and dose of treatment. They should use appropriate statistical analysis, otherwise this paper cannot be considered for publication as I cannot evaluate whether the effects they reposted are significant or not. A solution could also be to only focus on 1 treatment dose (10 mg/kg), which is the only one that seems to have an effect from their data.

Reply: There are two variables based on the experimental design (Control vs LPS). In first variable, the animal groups received aliskerin treatment at different doses. Later the groups received saline because LPS was dissolved in saline. In LPS variable, the animal groups, received aliskerin treatment at different doses followed with LPS administration. This experimental design was chosen to show the effect of aliskerin at basal level and against LPS-induced depressive-like behavior. We have chosen three different doses of aliskerin to examine the dose dependent of Aliskiren.

Re-reply: The 2 variables are not control vs LPS, but rather LPS AND Aliskiren treatment. As they use different doses, they introduce a third variable (the dose of aliskiren); thus, they should perform a three-way ANOVA. Again, they should use appropriate statistical analysis, otherwise this paper cannot be considered for publication as I cannot evaluate whether the effects they reposted are significant or not. 

Query 13: Additional experiments required: Survival curve, peripheral levels of inflammatory cytokines, temperature and body weight (sickness behavior). Moreover, the authors should also check or at least consider the stress induced to the mice by performing the forced swim and the tail suspension test 1 hour apart. CORT levels should be checked.

Reply: Authors aren thankful for these suggestions to the reviewers, further studies will be performed when next experimental protocol will be designed.

Re-reply: these additional experiments should be included in the context of this work as they are pivotal to claim their conclusions.

Author Response

Query 2. Line 53: minocycline is not a selective inhibitor of microglia.

Reply: Authors appreciate critical evaluation of manuscript and recommendation to improve the manuscript unto its excellent form. On the basis of previous research reports (K. Kobayashi and colleagues, 2013, cell death and disease, 4, e525) it has been conferred undoubtedly that minocyclin is selective inhibitor of microglia and prevent proinflammatory cascade of reaction.

Re-reply: For clarification, minocycline is a well-known antibiotic with anti-inflammatory, immunomodulatory and neuroprotective effects. The latters include inhibitory effects on the activities of key enzymes, like iNOS (Amin et al., 1997), inhibition of caspase-1 and caspase-3 activation (Chen et al., 2000), anti-apoptotic effects (Wang et al., 2003; Domercq and Matute, 2004; Jordan et al., 2007), reduction of p38 MAPK phosphorylation (Corbacella et al., 2004), etc. Thus, although it has been shown that it also inhibits microglial polarization (a concept not accepted anymore in the field as microglia can exist in a variety of cell states), this effects could also be ascribed to its activity on neurons (which obviously might feed-back to microglia).

Response: Thank you for your suggestion. As per the suggestions updated in the introduction section.

Query 7: More information about the type LPS used (Bacteria, strain, what type of purification) should given as this can have a huge impact on the inflammatory reaction. Also, why did they chose a dose of 0.5 mg/kg?

Reply: Information about the LPS strain used has been provided in the manuscript and authors selected 0.5mg/kg dose of LPS on the basis of previous reports

Re-reply: The previous reports used to choose the LPS dose should be clearly stated in the manuscript.

Response: Thank you for your comment. We have cited the previous reports used to choose the LPS dose in the manuscript. 

Query 11: The authors state that they used a 2-way ANOVA. Is this appropriate? Did they check the normal distribution of their data? Also, they have more than 2 variables: LPS, treatment and dose of treatment. They should use appropriate statistical analysis, otherwise this paper cannot be considered for publication as I cannot evaluate whether the effects they reposted are significant or not. A solution could also be to only focus on 1 treatment dose (10 mg/kg), which is the only one that seems to have an effect from their data.

Reply: There are two variables based on the experimental design (Control vs LPS). In first variable, the animal groups received aliskiren treatment at different doses. Later the groups received saline because LPS was dissolved in saline. In LPS variable, the animal groups, received aliskiren treatment at different doses followed with LPS administration. This experimental design was chosen to show the effect of aliskiren at basal level and against LPS-induced depressive-like behavior. We have chosen three different doses of aliskiren to examine the dose dependent of Aliskiren.

Re-reply: The 2 variables are not control vs LPS, but rather LPS AND Aliskiren treatment. As they use different doses, they introduce a third variable (the dose of aliskiren); thus, they should perform a three-way ANOVA. Again, they should use appropriate statistical analysis, otherwise this paper cannot be considered for publication as I cannot evaluate whether the effects they

Response: Thank you for your comment. We have done appropriate statistical analysis with the help of stat expertise by using two-way ANOVA. In the study, third variable dose of aliskiren as a factor therefore we used two-way ANOVA.

Query 13: Additional experiments required: Survival curve, peripheral levels of inflammatory cytokines, temperature and body weight (sickness behavior). Moreover, the authors should also check or at least consider the stress induced to the mice by performing the forced swim and the tail suspension test 1 hour apart. CORT levels should be checked.

Reply: Authors are thankful for these suggestions to the reviewers, further studies will be performed when next experimental protocol will be designed.

Re-reply: these additional experiments should be included in the context of this work as they are pivotal to claim their conclusions.

Response: Thank you for your suggestion. We will plan for the further studies to perform the suggested additional experiments. We have limitation of this study protocol. Hence, we have not included  the suggested work in this manuscript.

We would like to thank the reviewers and editor for a careful and thorough reading of this manuscript and for the thoughtful comments and constructive suggestions, which have greatly helped to improve the quality of this manuscript. We sincerely hope that it would have now met up to your expectations.